# Convolutional Neural Network for Pansharpening with Spatial Structure Enhancement Operator

**Weiwei Huang** [1], **Yan Zhang** [1], **Jianwei Zhang** [2] **and Yuhui Zheng** [1,*]

[1] Engineering Research Center of Digital Forensics, Ministry of Education, Nanjing University of Information Science and Technology, Nanjing 210044, China; 20201249414@nuist.edu.cn (W.H.); 201883290477@nuist.edu.cn (Y.Z.)

[2] School of Mathematics & Statistics, Nanjing University of Information Science and Technology, Nanjing 210044, China; zhangjw@nuist.edu.cn

[*] Correspondence: zheng_yuhui@nuist.edu.cn

**Abstract:** Pansharpening aims to fuse the abundant spectral information of multispectral (MS) images and the spatial details of panchromatic (PAN) images, yielding a high-spatial-resolution MS (HRMS) image. Traditional methods only focus on the linear model, ignoring the fact that degradation process is a nonlinear inverse problem. Due to convolutional neural networks (CNNs) having an extraordinary effect in overcoming the shortcomings of traditional linear models, they have been adapted for pansharpening in the past few years. However, most existing CNN-based methods cannot take full advantage of the structural information of images. To address this problem, a new pansharpening method combining a spatial structure enhancement operator with a CNN architecture is employed in this study. The proposed method uses the Sobel operator as an edge-detection operator to extract abundant high-frequency information from the input PAN and MS images, hence obtaining the abundant spatial features of the images. Moreover, we utilize the CNN to acquire the spatial feature maps, preserving the information in both the spatial and spectral domains. Simulated experiments and real-data experiments demonstrated that our method had excellent performance in both quantitative and visual evaluation.

**Keywords:** pansharpening; convolutional neural network; spatial structure enhancement

## 1. Introduction

In remote sensing images, a panchromatic (PAN) image always includes high spatial resolution and low spectral resolution [1], which covers all bands of visible light ranging from 0.38 to 0.76 μm. On the other hand, a hyperspectral (HS) image involves hundreds of bands, covering visible light to infrared [2], usually with sufficient spectral information but limited spatial structure. A multispectral (MS) image can be considered the special condition of an HS image, and the number of bands is generally only a few to a dozen, which is much less than that of an HS image. Compared to PAN images, MS images contain more spectral information. In practical application, for the sake of gaining a high-spatial-resolution MS (HRMS) image, in many cases, we often fuse a low-spatial-resolution MS (LRMS) image and a PAN image with abundant spatial details (PAN), hence yielding an HRMS image with sufficient spectral information and distinct spatial structure. We call this process pansharpening [3]. With further research, the spatial–spectral combination in HS images and PAN images has been brought into the study of pansharpening [4]. After decades of rapid development, pansharpening has been widely adopted in some respects, for example, in image classification and image segmentation.

Over the past few decades, many scholars have come up with multifarious pansharpening methods. All the methods could be roughly classified into four sorts: (1) component substitution (CS)-based methods, (2) multiresolution analysis (MRA)-based methods,

(3) variational optimization (VO)-based methods, and (4) deep learning (DL)-based methods [5]. In CS-based methods, firstly, we project the bands of MS images into another fresh spectral component, and secondly, we replace the part on behalf of the spatial information with the corresponding part of the PAN image. Some representative methods include adaptive GS (GSA) methods [6], principal component analysis (PCA) methods [7], and Brovey transform (BT) methods [8]. MRA-based methods first resolve the PAN image and the LRMS image into series bandpass channels. Secondly, we infuse the high-frequency information into the high-frequency channels of the LRMS image matching. Examples of MRA-based methods consist of the decimated wavelet transform (DWT) [9], the generalized Laplacian pyramid with a modulation transfer function-matched filter (MTF-GLP) [10], and Smoothing Filter-based Intensity Modulation (SFIM) [11]. VO-based approaches aim at two core tasks: the establishment of objective function, and the settlement of optimization. The most commonly used methods for constructing objective function are observation models [12] as the foundation and sparse representation [13]. Generally, the objective function of VO-based methods can be summarized into three aspects: (1) the spectral consistency model, (2) the structural consistency model, and (3) the prior model. An iterative algorithm is usually used for optimization solution, among which the gradient descent algorithm [14] is the most commonly used.

However, traditional pansharpening methods face the following problems:

1.  Loss of spectral information. For example, in CS-based methods, IHS methods [6] aim to convert the original MS image from RGB space to IHS space, and then replace an intensity (I) component with high-resolution images or images to be fused that are obtained by different projection methods. In the IHS space, the correlation of the three components is relatively low, which enables us to deal with the three components separately. The I component mainly reflects the total radiation energy of ground features and its spatial distribution, which is manifested as spatial characteristics, whereas H and S reflect spectral information. For methods of PCA methods [7], in contrast to the IHS transform method, the image can be decomposed into multiple components. The first principal component containing contour information can be replaced, which effectively improves the spatial resolution of multispectral images. However, the low correlation between the replacement components also causes distortion of spectral bands.

2.  Artifacts in the spatial structure. Among the MRA-based methods, a pansharpening method based on multiresolution DWT [9] is becoming increasingly popular due to its superior spectral retention ability. Nevertheless, because of the secondary subsampling in wavelet decomposition, artifacts usually exist in fusion results.

3.  Reliance on prior knowledge. Among some methods that rely on prior model, the most core issue is that the solution of the model is heavily dependent on prior knowledge. Moreover, the construction of the energy functional treats the degradation process as a linear transformation. However, in actual scenes, the degradation from an HR image to an LR image is a nonlinear process, with the superposition of various noises and disturbances.

Since previous years, deep learning has become a major area of research in artificial intelligence. Deep learning utilizes a great quantity of data as input to ultimately obtain a model with powerful analysis and recognition capabilities. Its primary task is to build a deep neural network. Deep learning has the following characteristics: It uses a large number of parameters and a nonlinear framework to better deal with nonlinear problems; instead of manually extracting features, the data are automatically screened and high-dimensional features are automatically extracted. Based on these characteristics, in recent years, research and applications have shown that deep learning has basically replaced previous related technologies, and remarkable breakthroughs have been made in image recognition and speech recognition.

Recently, because of the evolution of artificial intelligence and deep learning, convolutional neural networks (CNNs) have started to excel in super-resolution, and pan-

sharpening can be considered a super-resolution task, because both aim to enhance image resolution. Therefore, CNN has also been introduced to solve the problem with pansharpening. A super-resolution network with a three-layer CNN (SRCNN) was proposed by Dong et al. [15]. Based on this thread, Masi et al. first came up with a pansharpening method using CNN (PNN) [16], which applied the SRCNN model directly to the pansharpening task, achieving impressive performance. After that, a residual structure DiCNN [17] was introduced into pansharpening by He et al. As far as we know, DiCNN provided clear physical interpretation; meanwhile, it realizes rapid convergence while obtaining excellent-quality fusion results. However, the spatial structure knowledge of the images that cannot be completely exploited becomes the limitation of the DiCNN.

Among this research, we came up with an original pansharpening method, with the basic three-layer CNN as the foundation. Instead of the previous CNN architecture, the proposed pansharpening method utilizes the Sobel operator as a spatial structure enhancement operator to remove the low-frequency interference, which better acquires abundant spatial structure knowledge. The major contributions are as follows:

1.  We proposed an original pansharpening method that combines an edge-detection operator with a CNN to boost higher performance of edge feature extraction. The method that we came up with achieved preferable performance in simulated experiments and real-data experiments.
2.  We raised a three-layer CNN that can acquire spatial feature maps while preserving the spatial and spectral domain information.
3.  We designed a spatial structure enhancement operator that uses the Sobel operator as an edge-detection operator to gain high-frequency knowledge of the original PAN and LRMS images, hence obtaining the spatial features of the images.

## 2. Background

### 2.1. CNN Model

Feature extraction is a significant task for many deep learning problems. An essential problem with deep learning is to automatically combine simple features into more complex features. In the issue of deep learning, convolutional neural networks (CNNs) can be dated back several decades. CNN is a feedforward neural network used for feature extraction, which usually contains an input layer, convolutional layers, activation layers, pooling layers, and a full connection layer. It is a neural network in which a convolution operation replaces the traditional matrix multiplication operation. In recent years, CNNs have shown superior performance in image classification, target tracking, and face identification and other domains of computer vision. Common network structures include LeNet [18], VGGNet [19], and ResNet [20]. Thanks to the continuous development of modern techniques, CNNs have efficient implementations, with fast training achievable with high-speed and powerful GPUs. They have easily accessible data online, which is essential for training complex networks. Figure 1 shows a basic three-layer neural network structure.

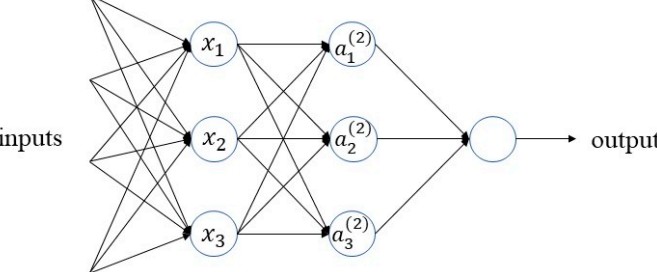

**Figure 1.** Three-layer neural network structure.

### 2.2. Pansharpening Based on CNN

Traditional methods possibly pose limitations in maintaining spectral information and spatial structure of remote-sensing images. The linear model is not suitable for the actual observation model in these traditional methods, particularly when the band coverage of the input PAN and LRMS images is incompletely overlapped, resulting in a highly nonlinear fusion process. In addition, in VO-based methods, the establishment of the objective function heavily depends on prior knowledge; meanwhile, these models are not strong in robustness for images with degraded quality. Furthermore, the optimization solution generally requires huge computational resource, which is time-consuming and may result in occasional errors.

To overcome these shortcomings, we exploited the generalization capabilities of the CNN, which consists of multiple layers. Through a multilayer convolutional neural network, a highly nonlinear integral transformation was finally formed. The most prominent advantage of the CNN model is that under the supervision of training samples, the whole parameters contained in the model can be updated, thereby reducing the reliability of prior knowledge and obtaining more accurate fitting accuracy.

CNNs are already widely applied in computer vision, for example, super-resolution. Pansharpening can be considered an extraordinary form of super-resolution. Dong et al. first raised the super-resolution method based on CNN (SRCNN) [15]. Borrowing from this model, Masi et al. raised a fundamental CNN model for pansharpening (PNN) [16] that applied the SRCNN model directly to the pansharpening task, achieving impressive performance. Then, Wei et al. improved the network structure by introducing a residual structure and subsequent dimensionality reduction (DRPNN) [21]. To solve the problem of poor interpretability in PNN and mismatching of residual dimensions in DRPNN, He et al. subsequently proposed DiCNN [17]. Figure 2 shows the network structure of the above-mentioned three CNN-based methods.

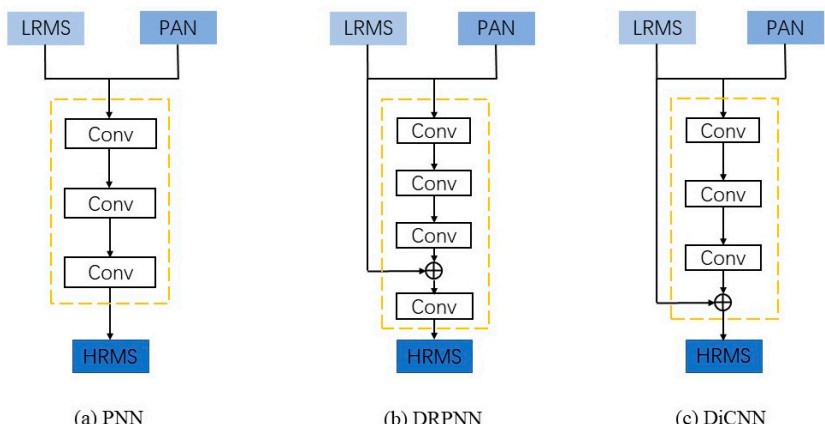

(a) PNN          (b) DRPNN          (c) DiCNN

**Figure 2.** Structures of the CNN-based methods.

### 2.3. Edge Detection Operator

Edge detection is a very significant approach to image processing in computer vision. It plays a significant part in image segmentation and target detection. Edge detection aims at discovering the collection of pixel points with drastic brightness changes in the image, which is often shown as contours. Once the edge of the image can be exactly measured and located, it indicates that the area, the diameter, and the shape of the practical object can be measured and located.

The basic methods of edge detection can be divided into three classes: (1) the edge operator of the first derivative. We use the convolution template to carry out the convolution operation for each pixel; afterwards, we select a suitable threshold to acquire the edge information according to the result. Common operators consist of the Roberts operator [22], Sobel operator [23], and Prewitt operator [24]. (2) Edge operator of the

second derivative. Based on the second derivative zero crossing, the common Laplacian operator [25] is noise sensitive. (3) Other edge operators. The first two types are all used to detect image edges through differential operators, as well as the Canny operator [26], which is an edge-detection optimization operator derived under certain constraints. We used the Sobel operator as the edge-detection operator in this paper.

The Sobel operator is a first-order differential operator primarily used in edge detection. Gaussian smoothing and differential methods are applied to compute the proximate gradient of the image. Using the Sobel operator at any pixel point will generate its relevant gradient vector.

## 3. The Proposed Methods

### 3.1. Pansharpening Methods

Traditional methods treat the space–spectrum fusion of remote-sensing images as a sick inverse problem. The degradation process of ideal high-resolution remote-sensing images is usually regarded as a linear model. We considered the expected HRMS images as $I$, $M$ denotes the LRMS images, and $P$ denotes the PAN images.

$$\begin{bmatrix} M \\ P \end{bmatrix} = \begin{bmatrix} DH \\ R \end{bmatrix} I + \begin{bmatrix} N_{MS} \\ N_{PAN} \end{bmatrix} \tag{1}$$

where $D$ represents the down-sampling matrix in the spatial domain, $H$ denotes a blurring matrix, $R$ is a spectral response matrix, and $N_{MS}$ and $N_{PAN}$ denote the additive noise.

The energy functional is mainly composed of three parts: the spectral consistency model, the spatial–structural consistency model, and the prior model. Most state-of-the-art methods regard fusion as minimizing the objective of the energy functional:

$$E(I) = \lambda_1 f_{spectral}(I, M) + \lambda_2 f_{spatial}(I, P) + \lambda_3 f_{prior}(I) \tag{2}$$

where $\lambda_i(i = 1, 2, 3)$ denotes the weights of the three items, respectively, $f_{spectral}(\cdot)$ represents the spectral fidelity model, $f_{spatial}(\cdot)$ represents the structural consistency model, and $f_{prior}(\cdot)$ represents the prior model, where logical hypothesis and prior knowledge are the foundation.

For the spectral fidelity model, many methods are defined:

$$f_{spectral}(I, M) = \sum_{b=1}^{B} \| DHI - \uparrow M \|_2^2 \tag{3}$$

where $\uparrow M$ indicates an up-sampling LRMS image and $B$ denotes the total bands of the LRMS image.

For the spatial–structural consistency model, the first variational method P+XS [27] is defined as follows:

$$f_{spatial}(I, P) = \| \sum_{b=1}^{B} \omega I - P \|_2^2 \tag{4}$$

where $\omega$ denotes a B-dimensional probability weight vector and $B$ represents the total bands of the LRMS image.

On account of the extensive application of CNNs in recent years, we can employ a simple CNN architecture to study the nonlinear mapping relationship among input $(P, M)$ and output $I$:

$$E(I) = \| f(P, M) - I \|_F^2 \tag{5}$$

where $f(\cdot)$ denotes the process of pansharpening. Although this simple CNN structure yields better results, it does not make full utilization of more spatial structure information of PAN images.

### 3.2. Network of Feature Extraction

In fact, it was empirically shown that a deeper CNN tends to extract more accurate image details and abundant structural information. However, due to the vanishing gradient problem, a deeper network cannot be fully learned. Therefore, in this research, we

adopted the three-layer CNN as our network model. Figure 3 is the global architecture of our method.

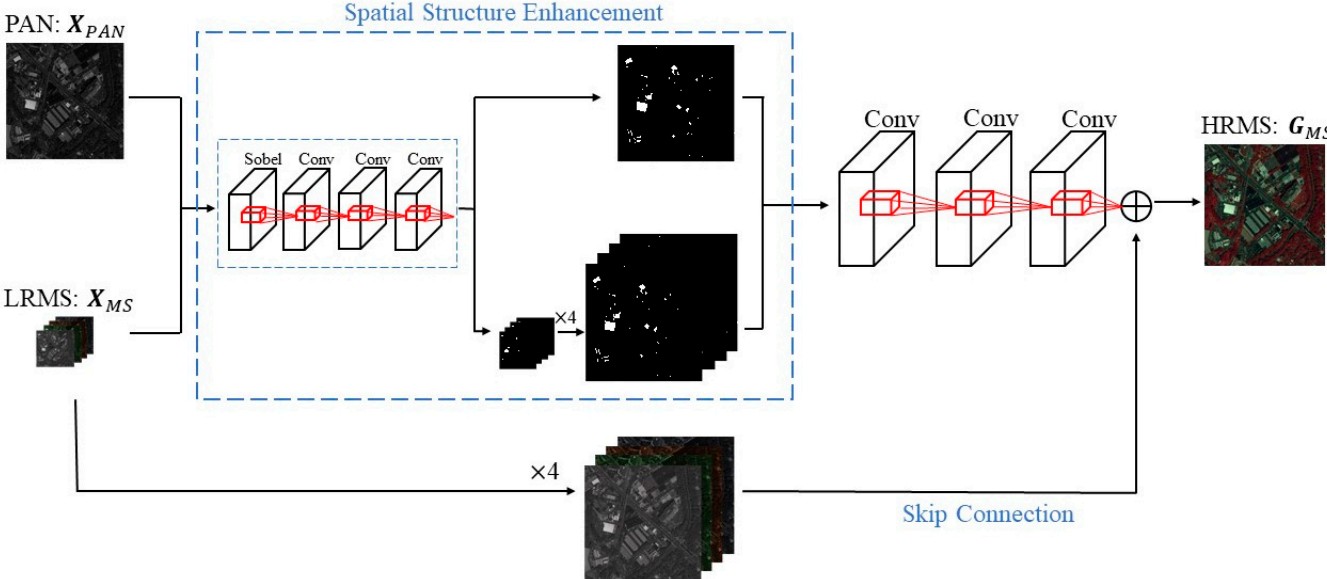

**Figure 3.** Architecture of the proposed method.

First, the input PAN and LRMS images were filtered with the spatial structure enhancement module to acquire the feature maps. Then we pre-interpolated the feature maps of the MS image, matching the size to a PAN image. Finally, our pansharpening method used a three-layer CNN and utilized skip connection to obtain the HRMS image. Specifically, the feature maps of the input image were obtained as input through the spatial structure enhancement module, and then the feature maps were superimposed and projected onto the three-layer CNN to obtain the filtered feature maps. The final output HRMS was obtained by adding the pre-interpolated LRMS and the filtered feature map through skip connection. Figure 4 shows the structure of our network.

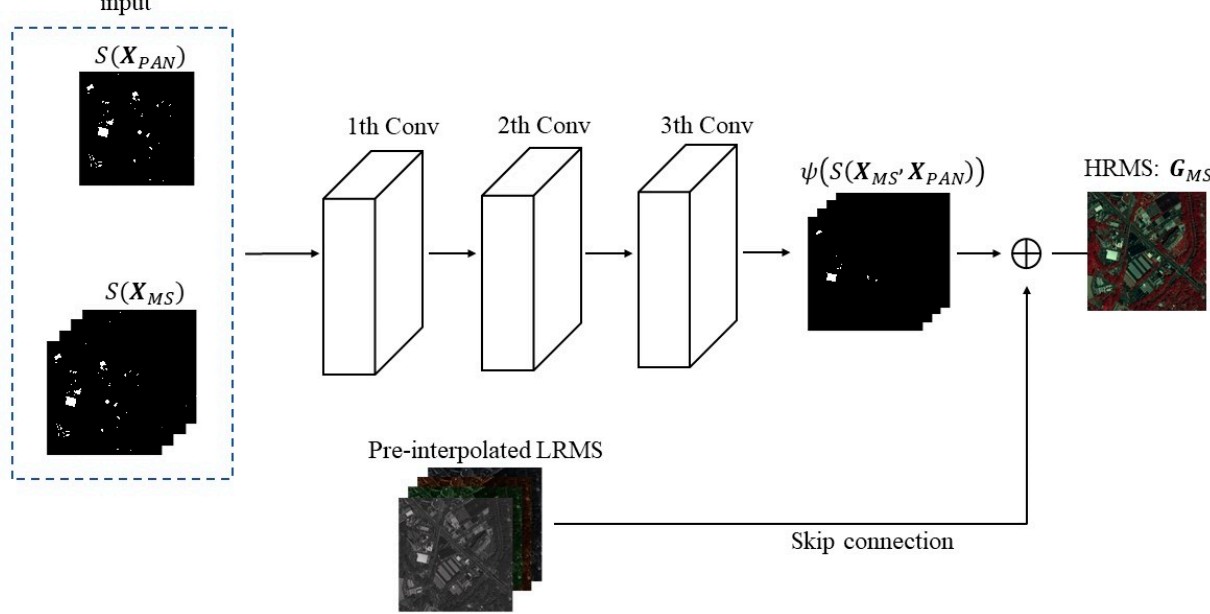

**Figure 4.** Flowchart of our network structure.

Equation (6) denotes the layer of feature extraction. $I$ refers to the input image of PAN and pre-interpolated LRMS, $*$ represents the convolution operation, and $W_i$ and $B_i$ are the weight and bias, respectively. Equation (7) denotes the residual layer, whereas Equation (8) represents the output layer.

$$L_i = \max(0, W_i * I + B_i), \ i = 1, 2, 3 \tag{6}$$

$$L_4 = W_4 * L_3 + B_4 \tag{7}$$

$$L_{output} = W_5 * [I + L_4] + B_5 \tag{8}$$

### 3.3. Spatial Structure Enhancement

For the sake of spatial structure enhancement, our method extracted high-frequency information using the Sobel operator. The Sobel operator is a kind of common edge-detection operator of the first-order derivative. In practical applications, the efficiency of the Sobel operator is higher than that of the Canny operator, but the edge is not as accurate as Canny. However, on many occasions, Sobel was the first choice, especially when the efficiency was important, and the fine texture was not too much of a concern. Sobel is usually directional; it can detect only horizontal edges or vertical edges, or both. The operator uses two $3 \times 3$ matrices to convolve with the original image, obtaining the gradient values of horizontal $G_x$ and vertical $G_y$, as shown in Equations (9) and (10), respectively. Finally, the approximate gradient $G$ is obtained by combining the above two results. A point is assumed to be an edge point if its gradient value is above a certain threshold.

$$G_x = \begin{bmatrix} -1 & 0 & +1 \\ -2 & 0 & +2 \\ -1 & 0 & +1 \end{bmatrix} * I \tag{9}$$

$$G_y = \begin{bmatrix} -1 & -2 & -1 \\ 0 & 0 & 0 \\ +1 & +2 & +1 \end{bmatrix} * I \tag{10}$$

$$G = \sqrt{G_x{}^2 + G_y{}^2} \tag{11}$$

In the spatial structure enhancement model, the input PAN image and the LRMS image are filtered by the Sobel operator, go through three convolution layers, and finally contain the feature map with rich spatial structure information. Figure 5 shows the architecture of the spatial structure enhancement model.

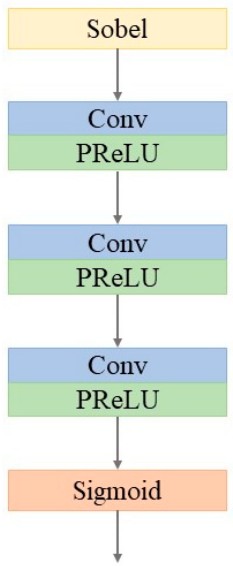

**Figure 5.** Architecture of the spatial structure enhancement model.

*3.4. Optimization Method*

In the proposed method, we employed the mean square error (MSE) function [28] as the objective function. Our objective function is described below:

$$LOSS = \frac{1}{N} \sum_{i=1}^{N} \| g_{mix}\left(X_{PAN}^{(i)}, X_{MS}^{(i)}\right) - F^{(i)} \|^2 \tag{12}$$

where $i$ is the index of patches; $N$ represents the sum of entire patches ; $X_{PAN}^{(i)}$ and $X_{MS}^{(i)}$ represent the $i$th patch for input PAN image and pre-interpolated MS image pairs, respectively; $g_{mix}$ refers to the image fusion operation; and $F^{(i)}$ represents the ground truth of the HRMS image.

For the sake of obtaining the HRMS image, we devoted time to adjusting the weight and bias to make the overall loss smaller, and to obtaining the parameters of each neuron corresponding to the minimum value of the overall cost (i.e., the weight and bias). Here, we used stochastic gradient descent (SGD) [29] to optimize the objective function, updating the parameters with following formula:

$$\begin{aligned} W^{(l)} &= W^{(l)} - \mu \frac{\partial LOSS_{total}}{\partial W^{(l)}} \\ &= W^{(l)} - \frac{\mu}{N} \sum_{i=1}^{N} \frac{\partial LOSS_{(i)}}{\partial W^{(l)}} \end{aligned} \tag{13}$$

$$\begin{aligned} b^{(l)} &= b^{(l)} - \mu \frac{\partial LOSS_{total}}{\partial b^{(l)}} \\ &= b^{(l)} - \frac{\mu}{N} \sum_{i=1}^{N} \frac{\partial LOSS_{(i)}}{\partial b^{(l)}} \end{aligned} \tag{14}$$

where $LOSS_{total}$ represents the whole (average) loss of entire training data; $W^{(l)} \epsilon \mathbb{R}^{n_l \times n_{l-1}}$ and $b^{(l)} = \left(b_1^{(l)}, b_2^{(l)}, \ldots, b_{nl}^{(l)}\right)^{\mathrm{T}} \epsilon \mathbb{R}^{n_l}$ represents the weight matrix and bias from layer $l-1$ to layer $l$, respectively; and $LOSS_{(i)}$ is the loss of single training data.

We summarize specific implementation of our pansharpening method in Algorithm 1.

---

**Algorithm 1.** Spatial structure enhancement convolution operator pansharpening algorithm.

---

**Input:** LRMS image $X_{MS}$;
       PAN image $X_{PAN}$
**Output:** HRMS image $G_{MS}$
1: **for** each $patch_{i,j} \in X_{MS}, X_{PAN}$ **do**
2: Extract feature maps $S'(X_{MS})$ of $patch_i$ by Sobel operator
3: Extract feature maps $S(X_{PAN})$ of $patch_j$ by Sobel operator
4: Up-sampling $S'(X_{MS})$ four times by bicubic interpolation, generate the feature maps $S(X_{MS})$
5: Combine feature maps generated after Step 3 and Step 4, generate the final feature maps
      $S(X_{MS}, X_{PAN})$
6: Filter $\psi(S(X_{MS}, X_{PAN}))$ the feature maps $S(X_{MS}, X_{PAN})$ by three-layer convolutional neural
      network
7: Fuse the pre-interpolated LRMS with the final feature maps, generate the high- resolution
      image $G_{MS}$ after fusion
8: Evaluate the $LOSS$ of the ground truth and the fused image $G_{MS}$
9: **if** $LOSS < \tau$
10: Return to Step 2
11: **end if**
12: **end for**

---

## 4. Experiments and Discussion

*4.1. Experimental Settings*

4.1.1. Datasets

We conducted experiments using a dataset from the Worldview-2 satellite, which included a PAN image in which sensor bands ranged from 0.45 μm to 0.8 μm, and an MS

image with eight bands. The Worldview-2 sensor is the primary high-resolution eight-band multispectral sensor capable of providing panchromatic images at 0.46 m and multispectral images at 1.8 m resolution. Except for four normal bands (red, green, blue, and near-infrared 1), Worldview-2 also includes four supernumerary bands, providing users with the capacity to perform accurate change detection and mapping.

Meanwhile, we used a dataset from the Quickbird sensor that provided a PAN image formed from wavelengths of 450 nm to 900 nm. There are four standard bands of a multi-spectral image. At present, Quickbird satellite data have been widely used in land–water resource surveys, land resource surveys, vegetation resource surveys, geological surveys, urban remote-sensing surveys, archaeological surveys, and environmental monitoring.

### 4.1.2. Parameters and Training

For our experiment, we split datasets into 90% for training and 10% for validation, and designated the batch size to 32 and the patch size to 64. The training process of our network spanned 300 epochs. We used the objective function in Equation (12) of SGD minimization. Our initial learning rate was $10^{-3}$. Then we multiplied by $10^{-1}$ at $10^5$, $2 \times 10^5$ iterations, and finally, we terminated the process at $3.0 \times 10^5$ iterations. In the spatial structure enhancement module, we set the convolution sizes to $s_1 = s_2 = s_3 = 3$ and the convolution numbers to $a_1 = a_2 = 32$.

With the Tensorflow1.13 deep learning framework, the CNN-based methods used a GPU (Nvidia GeForce RTX 2080 8 GB with CUDA 10.1 and CUDNN 7.4) for training through the Windows 10.0 operating system; then the Tensorflow deep learning framework on Python was tested using a CPU (desktop with Intel I5 and 16 GB RAM) on Windows 10.

### 4.2. Assessments

#### 4.2.1. Quantitative Assessments

Five quantitative performance measures were used to evaluate performance in the simulated experiments: the spectral angle mapper (SAM) [30], relative dimensionless global error in synthesis (ERGAS) [31], the spatial correlation coefficient (SCC) [32], the peak signal–noise ratio (PSNR) [33], and the mean structural similarity (SSIM) [33]. There are three non-reference numeric metrics to assess real-data experiments: the quality with no-reference index (QNR) [34] and its spatial and spectral components ($D_S$ and $D_\lambda$) [35].

In addition, we compared our method to five representative methods: PCA [7], MTF-GLP [10], SFIM [11], PNN [16], and DiCNN [17]. The latter two methods are CNN-based methods.

#### 4.2.2. Visual Assessments

Although accurate measures can be estimated by the quality of the fusion outcomes, subjective visual assessment is still required to detect any significant distortions in the fused images, which may not be demonstrated in the quality assessment. Therefore, all of the experimental results are displayed as true color.

### 4.3. Experiment 1: Simulated Experiments

The PAN image and the LRMS image were downsampled by Gaussian function to obtain a low-resolution image at first in these experiments. After that, the MS image, which had the same size as the PAN image, was interpolated using a polynomial kernel (EXP) [36]. We employed the processed images as the input of the network.

Table 1 shows the quantitative results of the Worldview-2 images. By comparing the results, it can be clearly observed that most indices of our methods performed the best. As we can observe, the evaluation indices of traditional methods were mostly lower than those of CNN-based methods, and three CNN-based methods received a huge boost in performance, which proves the excellence of CNN. Moreover, most of the evaluation indices of our method were not only much higher than traditional methods, but also better than those of other CNN-based methods. This indicates that the spatial structure enhancement

operator proposed by us takes full advantage of the structural texture of images, thereby improving the performance indices, which also fully illustrates the outstanding ability of the proposed method.

**Table 1.** Quantitative results of the Worldview-2 dataset.

|          | ERGAS (↓) | SAM (↓) | SCC (↑) | PSNR (↑) | SSIM (↑) | Time (s) |
|----------|-----------|---------|---------|----------|----------|----------|
| EXP      | 5.9835    | 2.8785  | 0.5988  | 20.7926  | 0.7092   |          |
| PCA      | 12.484    | 2.5433  | 0.6486  | 23.9662  | 0.7284   | 0.54     |
| SFIM     | 5.8498    | 2.5448  | 0.8481  | 24.1771  | 0.7132   | 0.79     |
| MTF-GLP  | 3.8994    | 2.5426  | 0.9139  | 27.3228  | 0.8021   | 0.47     |
| PNN      | 3.9079    | 1.5470  | 0.9071  | 28.9181  | 0.8048   | 1.99     |
| DiCNN    | **3.6055**| 1.5457  | 0.9365  | 29.7285  | 0.8638   | 2.03     |
| Proposed | 3.6246    | **1.5455** | **0.9377** | **30.1652** | **0.8679** | 2.45 |

Figure 6 displays the total fusion images in the Worldview-2 dataset. Figure 6a is the ground truth of the LRMS image. As we can observe, there was significant spectral distortion in the PCA method, as shown in Figure 6b. According to the results, in the SFIM methods, there were apparent artifacts in the processed HRMS image in Figure 6c. Although the MTF-GLP method maintained many spectral features, the spatial information was limited, as shown in Figure 6d. Figure 6e–g proves the excellent expression of the three CNN-based methods; however, by comparison, our proposed method not only kept good spectral consistency of the MS image, but also obtained richer structure details than other CNN-based methods. For example, the white line in the lower right corner is clearer than the other results. This benefitted from our spatial structure enhancement operator, which extracted the exquisite texture information to enhance the expression of the deep features of the image, and finally reflected more precise edge characterization in the fused image.

Traditional methods have relatively less computational time loss because they do not have to train abundant parameters of a deep neural network. Compared to traditional methods, the toughest part of the pansharpening methods based on CNN is the training of CNN model parameters. In general, the deeper the layers, the longer the calculation time. Therefore, CNN-based methods consume more computational cost. Table 1 displays the average run time (in seconds) for a single 256 × 256 image in the Worldview-2 dataset.

Similarly, in the case of the Quickbird dataset, our method obtained a substantial improvement over the traditional methods, as shown in Figure 7. By comparison, it is obvious that the results were the same as those shown in Figure 6. There was a distinct disadvantage of the spectrum in the final result in the PCA method. Other traditional methods need to be improved in terms of detail preservation. Among the three CNN-based methods, our method maintained better spatial and spectral information.

According to our analysis, that there may be potential for further pansharpening improvement by combing the edge extraction operator with the CNN structure.

### 4.4. Experiment 2: Real-Data Experiments

To accomplish the expected HRMS image, we imported the initial PAN and LRMS image straightly into the model without any preprocessing among the experiments.

Table 2 shows the non-reference assessment results of the Quickbird dataset. Although the indices of $D_S$ and $D_\lambda$ were better in the DiCNN method, the proposed method retained similar numerical values. For the QNR index, the method we proposed achieved the best performance. These non-reference evaluation indicators represent the restoration quality of direct fusion in the absence of ground truth, which also indicates that our method can achieve great results in practical situation of directly integrating images.

**Table 2.** Quantitative results of the Quickbird dataset.

|  | QNR ($\uparrow$) | $D_S$ ($\downarrow$) | $D_\lambda$ ($\downarrow$) |
|---|---|---|---|
| PCA | 0.6257 | 0.2679 | 0.1454 |
| SFIM | 0.8700 | 0.0523 | 0.0820 |
| MTF-GLP | 0.8166 | 0.0894 | 0.1033 |
| PNN | 0.8237 | 0.1287 | 0.2406 |
| DiCNN | 0.8606 | **0.1115** | **0.0313** |
| Proposed | **0.8675** | 0.1147 | 0.0452 |

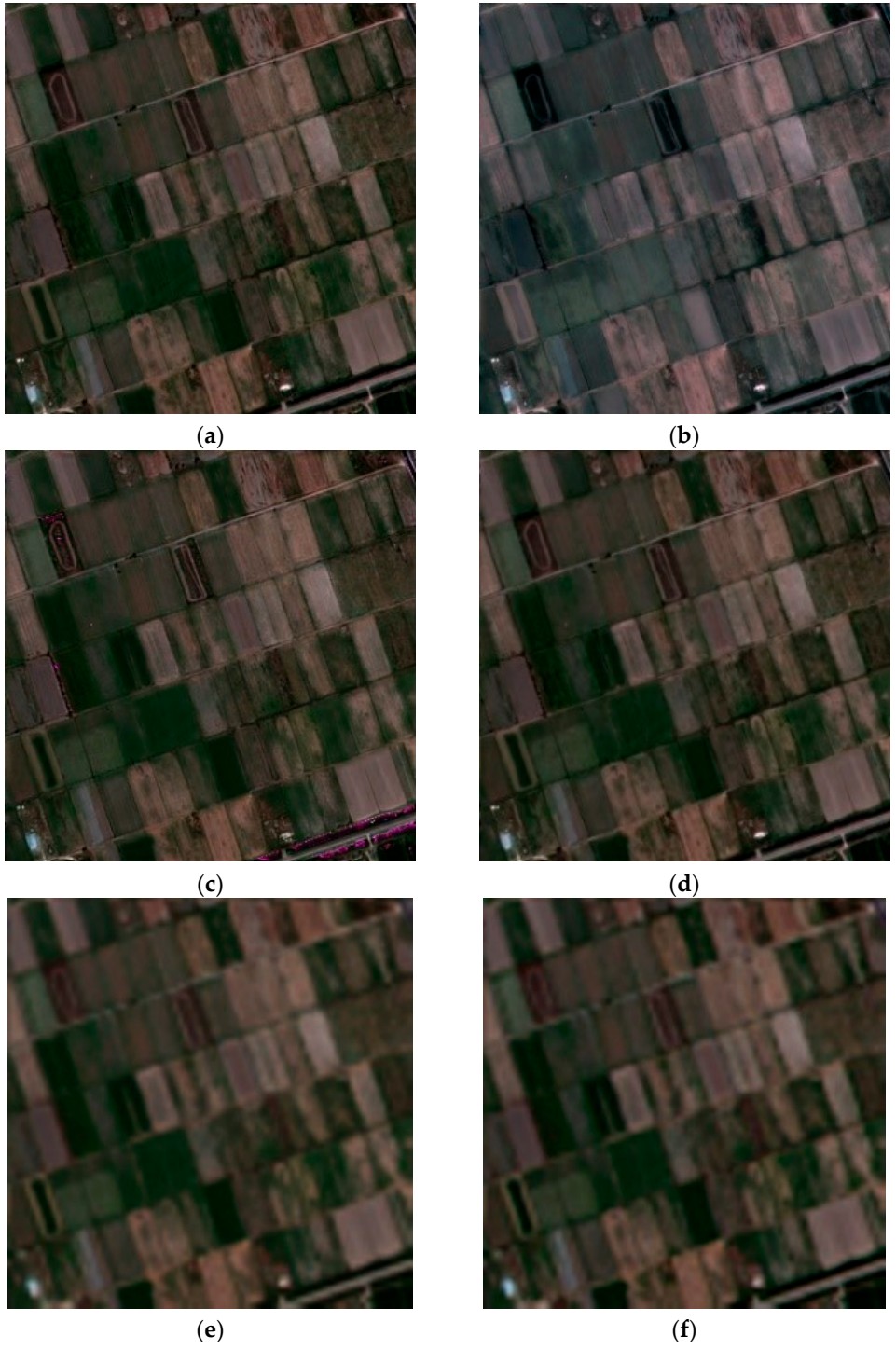

**Figure 6.** *Cont.*

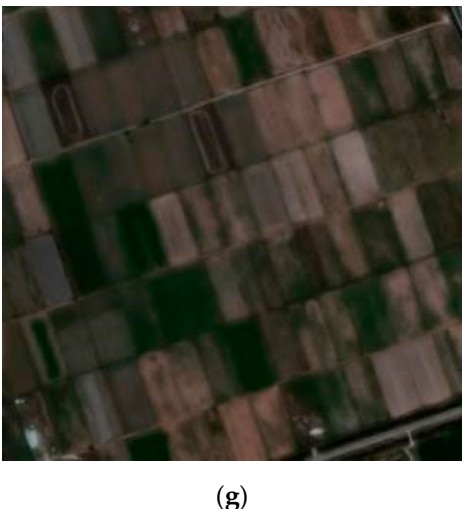

(**g**)

**Figure 6.** Simulated results for the Worldview-2 dataset. (**a**) Ground truth; (**b**) PCA; (**c**) SFIM; (**d**) MTF-GLP; (**e**) PNN; (**f**) DiCNN; (**g**) proposed.

A group of real-data experimental results of Quickbird images is shown in Figure 8. All of them are displayed in true color. Figure 8a is the original PAN image. In contrast to the representative methods, we can observe that the PCA methods could not maintain good spectral preservation in Figure 8b. There was a serious blur in the SFIM method, as displayed in Figure 8c. The performance of the MTF-GLP method was relatively good in traditional methods. Figure 8e–g are the visual exhibitions of three CNN-based methods. They obtained significantly more outstanding results than traditional methods. Especially in the proposed method, the fusion results are prominent, the spatial structure is more abundant, and the edges of cars and houses are more distinct. Due to the utilization of CNN, the fused image is smooth and detailed; meanwhile, the proposed spatial structure enhancement operator can excavate the deep edge features of the image based on CNN architecture. The combination of the two modules can finally restore the satisfactory image.

Figure 9 shows the real-data results of the Worldview-2 dataset. As displayed in Figure 9e–g, three CNN-based methods achieved more outstanding performance not only in the spatial domains but also in the spectral domains. Moreover, our approach better highlighted the edges of the house and the shoreline. The results presented in Figure 9 fully prove the superiority of our method.

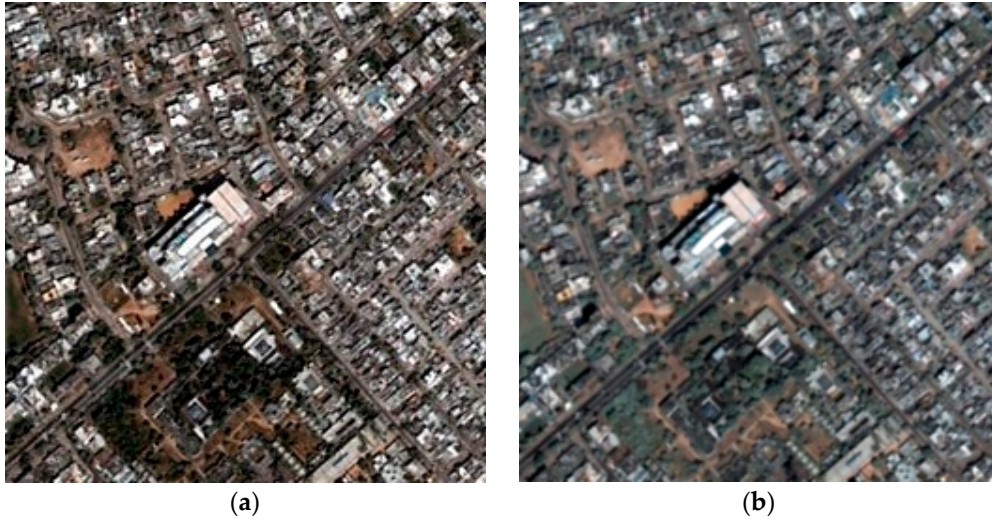

(**a**)                      (**b**)

**Figure 7.** *Cont.*

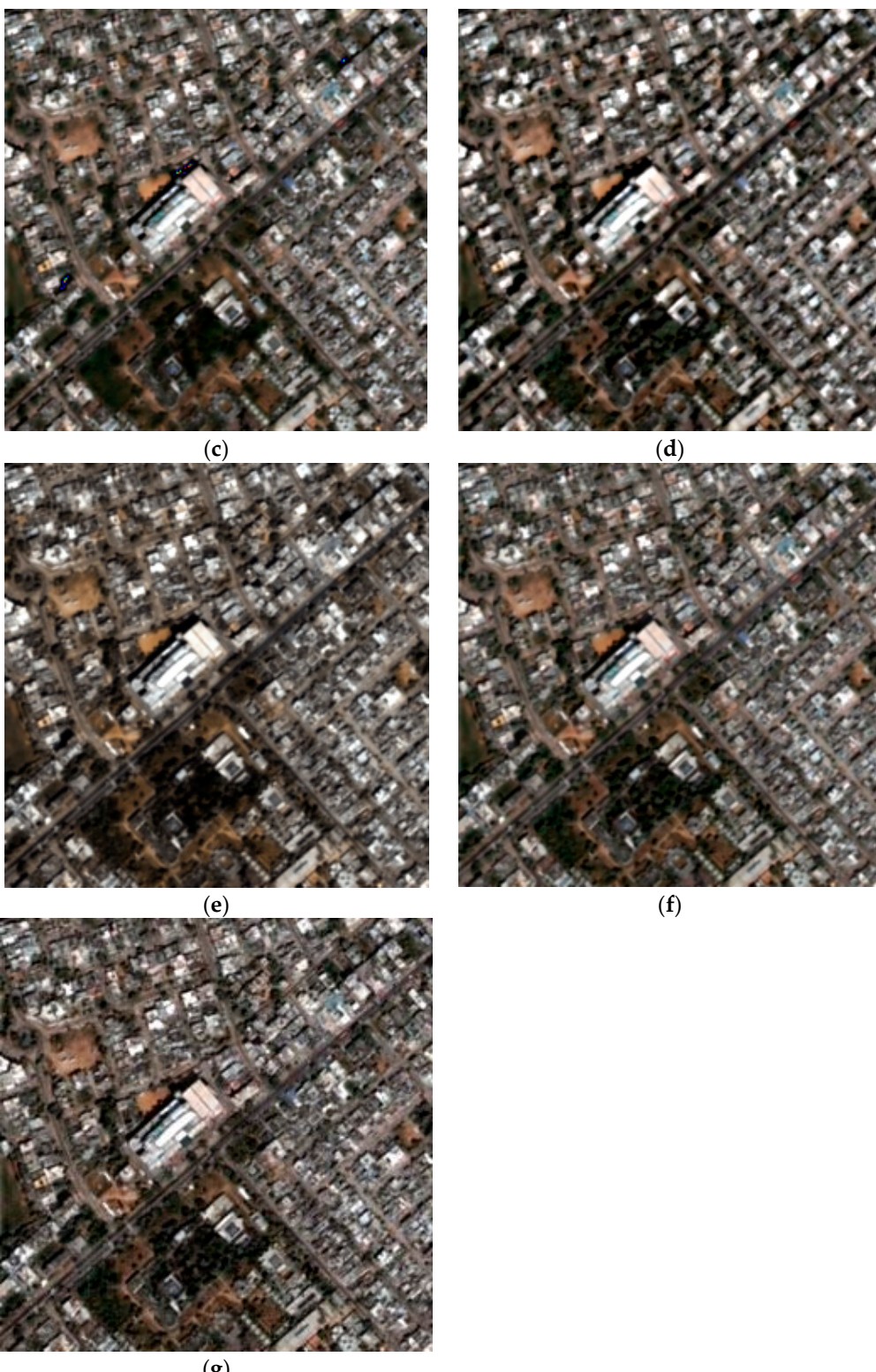

**Figure 7.** Simulated results of the Quickbird dataset. (**a**) Ground truth; (**b**) PCA; (**c**) SFIM; (**d**) MTF-GLP; (**e**) PNN; (**f**) DiCNN; (**g**) proposed.

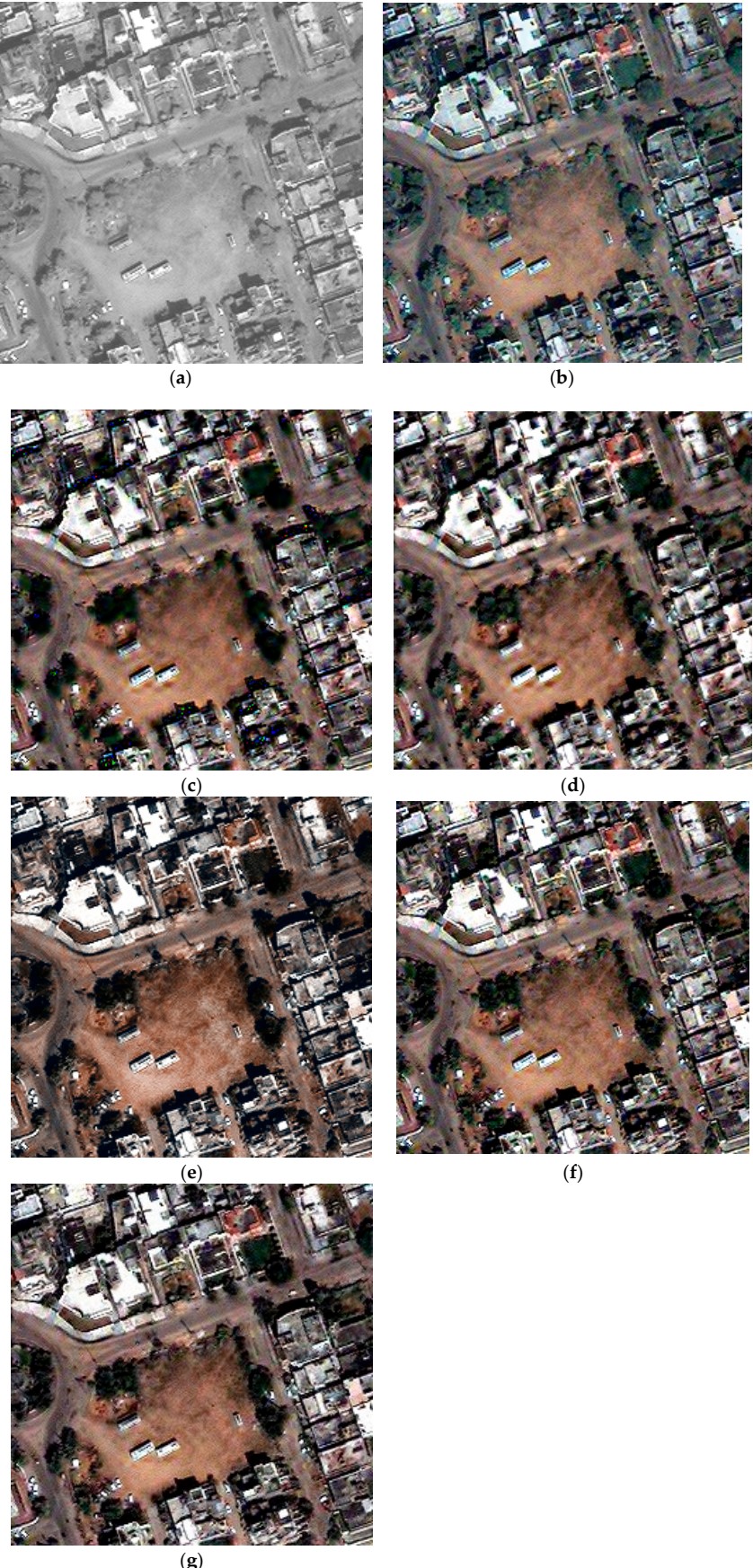

**Figure 8.** Real-data results for the Quickbird dataset. (**a**) Original PAN; (**b**) PCA; (**c**) SFIM; (**d**) MTF-GLP; (**e**) PNN; (**f**) DiCNN; (**g**) proposed.

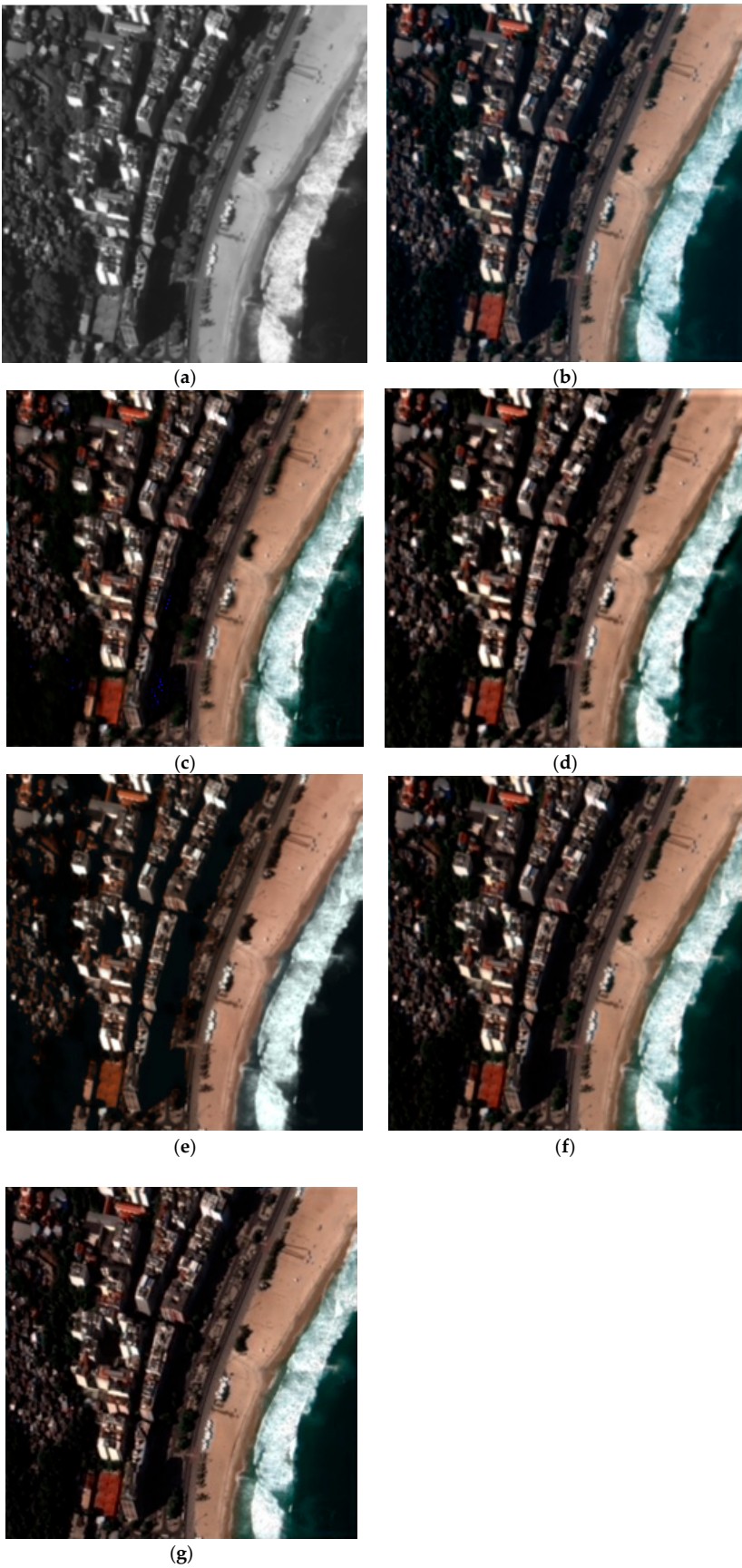

**Figure 9.** Real-data results for the Worldview-2 dataset. (**a**) Original PAN; (**b**) PCA; (**c**) SFIM; (**d**) MTF-GLP; (**e**) PNN; (**f**) DiCNN; (**g**) proposed.

## 5. Conclusions

An original pansharpening method with a spatial structure enhancement operator and convolutional neural network emerged in this research. It extracted high-frequency spatial information of LRMS and PAN images using the Sobel operator, utilizing CNN to achieve the excepted HRMS image. Unlike traditional methods, our method not only retained the spectral information, but also made maximum utilization of the spatial structure knowledge in input images.

Among our simulated and real-data experiments on the Worldview-2 and Quickbird datasets, the performance of the traditional methods was dissatisfactory, which may have caused the loss of spectral information and artifacts in the spatial structure. In general, deep learning (DL)-based methods perform better, which can maintain the consistency of the spectrum and spatial structure. Our experimental results in the visual and objective evaluation indicators achieved good performance, which proves the superior performance of our method.

Although the proposed pansharpening method has made considerable progress, there are still some shortcomings. For example, there is a lack of publicly available large datasets, most of the existing remote-sensing data involve national security, and data confidentiality is relatively strong. Secondly, we regard a deep learning model as a black box, as its learning procedure usually lacks interpretability. Due to the large computational loss, the existing deep learning models are relatively simple. In the future, we will extend to building large public datasets for research and explore the possibility of designing pansharpening networks with more hidden layers and more intricate interconnections between multiple convolutional layers.

**Author Contributions:** Conceptualization, W.H., Y.Z. (Yan Zhang), J.Z. and Y.Z. (Yuhui Zheng); methodology, W.H.; software, W.H.; validation, W.H.; formal analysis, W.H.; investigation, W.H. and Y.Z. (Yan Zhang); resources, W.H., J.Z. and Y.Z. (Yuhui Zheng); data curation, W.H.; writing—original draft preparation, W.H.; writing—review and editing, W.H. and Y.Z. (Yuhui Zheng); visualization, W.H.; supervision, W.H., J.Z. and Y.Z. (Yuhui Zheng); project administration, W.H.; funding acquisition, Y.Z. (Yuhui Zheng). All authors have read and agreed to the published version of the manuscript.

**Funding:** This work was supported in part by the National Natural Science Foundation of China under Grants U20B2065, 61972206 and 62011540407; in part by the Natural Science Foundation of Jiangsu Province under Grant BK20211539; in part by the 15th Six Talent Peaks Project in Jiangsu Province under Grant RJFW-015; in part by the Qing Lan Project; in part by the National Research Foundation of Korea under Grants NRF-2020K2A9A2A06036255 and FY2020]; in part by the PAPD fund.

**Institutional Review Board Statement:** Not applicable.

**Informed Consent Statement:** Not applicable.

**Data Availability Statement:** All data included in this study are available upon request by contact with the corresponding author.

**Conflicts of Interest:** The authors declare no conflict of interest.

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
