# Peer review of "Convolutional Neural Network for Pansharpening with Spatial Structure Enhancement Operator"

_remotesensing, doi:10.3390/rs13204062_

Round 1
Reviewer 1 Report
This paper proposes a novel pansharpening method combining a spatial structure enhancement operator with a convolutional neural network (CNN). This method uses the Sobel operator to extract the high-frequency information in PAN image and MS image. Then the CNN is used to acquire the spatial feature maps. Some experiments on real-data demonstrate the effectiveness of the proposed method. The main comments are
- Please provide more comprehensive reviews about Pansharpening, such as hyperspectral for Pansharpening (10.1109/MGRS.2015.2440094, 10.1109/JSTARS.2019.2917584, and 10.1109/TGRS.2020.2963848)
- The motivation should be explained why proposing this method.
- Please provide the detailed network structure of the proposed method.
- For the experimental results, please enhance the analysis about the merits of the proposed method.
- Please discuss the existing problems about the proposed method for future work.

Reviewer 2 Report
A novel pansharpening approach was introduced in this work, combining the convolutional neural network and a spatial structure enhancement scheme. In the spatial enhancement stage the authors employed the edge detection operator for extracting the structure details, cascaded with a three-layer convolutional network which plays a role of the trainable mapping. The proposed algorithm was evaluated on the WorldView-2 and QuickBird images. The experiments shows the superiority of the proposed method. The manuscript was well-organized. My concerns are as follows:
- It is better to add symbols used in algorithm 1 to each diagram in Fig. 3 and Fig. 4.
- In the spatial structure enhancement module, the edges were detected and extracted with the Sobel operator, then a three-layer CNN was employ to generate the feature maps. However, it is not clear how the parameters of the layers were set. Could the authors offer more detains on this issue?
- The performance related to the sole interpolation of the original LR MS image (typically indicated as EXP) was not reported in the tables. Please offer more details on it.
- The computational time of the proposed algorithm should be reported and analyzed in experiments.
- There are some minor grammatical problems in the manuscript. In addition, the English expressions should be carefully polished.

Reviewer 3 Report
The authors have done good research in proposing a novel pansharpening method with a spatial structure enhancement operator and convolutional neural network. The topic of this work is very interesting and with possible applicability. The paper is well written and has good scientific impact. However, there are some aspects that need clarification and improvements as follows:
- I think, it’s better to collect all abbreviations for this manuscript to one table.
- Fig. 3 shouldn’t be appears before indicates it, the same with all Figures and Tables.
- The authors do not refer to the number of feature dimension in the proposed method.
- I think Section 3 title should be rewritten as 3. Methods à 3. The Proposed Method, and 3.1. Review of Pansharpening Methods à 3.1. Pansharpening Method
- The authors have compared the proposed method with five representative methods: PCA [5], MTF-GLP [8], SFIM [10], PNN [22], and DiCNN [23], however, most of the previous references are old and there are no recently references that are utilized CNN such as:
- Vitale, Sergio, and Giuseppe Scarpa. "A detail-preserving cross-scale learning strategy for CNN-based pansharpening." Remote Sensing3 (2020): 348.
- Ozcelik, Furkan, et al. "Rethinking CNN-based pansharpening: Guided colorization of panchromatic images via GANS." IEEE Transactions on Geoscience and Remote Sensing4 (2020): 3486-3501.
- Figures 1 : 5 are not clearly, the authors should represent it with more contrast especially writing expressions.
- Furthermore, the manuscript has numerous grammatical mistakes. Check the manuscript carefully again.
- Little typos as:
- The expression DiCNN à DICNN.
- a1 = a2 = 32. à a1 = a2 = 32.
- s1 = s2 = s3 = 3
... etc
Round 2
Reviewer 1 Report
This paper proposes a convolutional neural network for pansharpening using spatial structure enhancement operator. This method uses the Sobel operator as an edge detection operator to extract abundant high-frequency information from the input images. Then, CNN is used to obtain the spatial feature maps. Some experiments demonstrate the effectiveness of the proposed method. The main comments are
- The abstract should be rewritten to highlight the motivation why proposes this method for pansharpening and enhance the innovation of the proposed method.
- For feature extraction, some traditional methods should be mentioned to highlight the merits of deep learning such as 10.3390/rs13071363 and 10.1109/TGRS.2021.3110855.
- In Fig. 3, the PAN image and the LRMS image have different scales how to input them into the network. Maybe, it needs to preprocess them for inputting. Please give more detailed information.
- For the experimental results, the discussion should be enhanced to highlight the importance of the proposed method. In addition, please analyze why the proposed can achieve better results.
Reviewer 3 Report
I can see the authors work hard to resolve all reviewers comments.
Thanks for the good work.
Author Response
Thank you very much for your sincere suggestions.
Round 3
Reviewer 1 Report
All my problems have been addressed.